# Structure and neutralization mechanism of a human antibody targeting a complex Epitope on Zika virus

Cameron Adams[1], Derek L. Carbaugh[1✪], Bo Shu[2,3✪], Thiam-Seng Ng[2,3], Izabella N. Castillo[1], Ryan Bhowmik[1], Bruno Segovia-Chumbez[1], Ana C. Puhl[4], Stephen Graham[1], Sean A. Diehl[5], Helen M. Lazear[1], Shee-mei Lok[2,3], Aravinda M. de Silva[1]*, Lakshmanane Premkumar[1]*

**1** Department of Microbiology and Immunology, University of North Carolina School of Medicine, Chapel Hill, North Carolina, United States of America, **2** Program in Emerging Infectious Diseases, Duke-National University of Singapore Medical School, Singapore, Singapore, **3** Centre for Bio-Imaging Sciences, Department of Biological Sciences, National University of Singapore, Singapore, Singapore, **4** Center for Integrative Chemical Biology and Drug Discovery, Chemical Biology and Medicinal Chemistry, Eshelman School of Pharmacy, University of North Carolina, Chapel Hill, North Carolina, United States of America, **5** Department of Microbiology and Molecular Genetics, University of Vermont Larner College of Medicine, Burlington, Vermont, United States of America

✪ These authors contributed equally to this work.
* aravinda_desilva@med.unc.edu (AMS); prem@med.unc.edu (LP)

**Data Availability Statement:** All the data for the crystal and cryoEM structures have already been deposited at PDB with the accession codes 8DV6

## Abstract

We currently have an incomplete understanding of why only a fraction of human antibodies that bind to flaviviruses block infection of cells. Here we define the footprint of a strongly neutralizing human monoclonal antibody (mAb G9E) with Zika virus (ZIKV) by both X-ray crystallography and cryo-electron microscopy. Flavivirus envelope (E) glycoproteins are present as homodimers on the virion surface, and G9E bound to a quaternary structure epitope spanning both E protomers forming a homodimer. As G9E mainly neutralized ZIKV by blocking a step after viral attachment to cells, we tested if the neutralization mechanism of G9E was dependent on the mAb cross-linking E molecules and blocking low-pH triggered conformational changes required for viral membrane fusion. We introduced targeted mutations to the G9E paratope to create recombinant antibodies that bound to the ZIKV envelope without cross-linking E protomers. The G9E paratope mutants that bound to a restricted epitope on one protomer poorly neutralized ZIKV compared to the wild-type mAb, demonstrating that the neutralization mechanism depended on the ability of G9E to cross-link E proteins. In cell-free low pH triggered viral fusion assay, both wild-type G9E, and epitope restricted paratope mutant G9E bound to ZIKV but only the wild-type G9E blocked fusion. We propose that, beyond antibody binding strength, the ability of human antibodies to cross-link E-proteins is a critical determinant of flavivirus neutralization potency.

## Author summary

Zika virus (ZIKV) is a mosquito-borne virus that infects people living in many tropical regions of the world. While most ZIKV infections are mild, the virus can cause serious

and 7YAR. All other relevant data are within the main manuscript and supporting information files.

**Funding:** This work was supported by the US Centers for Disease Control and Prevention (00HVCLJB-2017-04191 to AMdS), the US National Institutes of Allergy and Infectious Diseases (R01AI107731 to AMdS; R21AI144631 to HML; T32 AI007419 to DLC; U01AI141997 to SAD), the US National Institute of General Medicine (P20GM12549 to SAD) and Singapore National Research Foundation Competitive Research Project (NRF-CRP17-2017-04 to SmL). The Duke-NUS Signature Research Programme was supported by the Ministry of Health, Singapore (PI: SmL). The funders had no role in study design, data collection and analysis, decision to publish, or preparation of the manuscript.

**Competing interests:** The authors have declared that no competing interests exist.

neurological problems and infection during pregnancy can lead to severe fetal birth defects. People infected with ZIKV develop antibodies that assist with viral clearance as well as protection from repeat infections. Previously, we isolated a ZIKV neutralizing and protective antibody from a person who had recovered from a ZIKV infection. To understand how this antibody neutralized the virus, we solved the structure of the antibody bound to ZIKV and performed experiments with mutated viruses and antibodies. Our studies indicate that the antibody blocks infection by cross-linking and paralyzing proteins on the viral surface required for the virus to enter cells.

## Introduction

Zika virus (ZIKV) is one of several medically important flaviviruses transmitted by mosquitos to humans [1–4]. Flaviviruses are positive-sense RNA viruses of approximately 50 nm in diameter with a lipid bilayer containing the envelope (E) and premembrane/membrane (prM/M) glycoproteins [5]. The ectodomain of E protein is composed of two non-continuous domains (EDI and EDII) and a continuous immunoglobulin-like domain (EDIII). Processing of prM into M and pH-induced E protein conformational rearrangement occurs during virus egress. In the mature infectious virion, E monomers form stable head-to-tail homodimers and 90 E homodimers are packed to create an icosahedral particle with a smooth protein surface [6].

The flavivirus E glycoprotein, essential for viral attachment and entry into host cells, is a major target of human antibodies [6]. Some anti-E protein antibodies are strongly neutralizing and protective, while others are weakly neutralizing and implicated in enhanced viral replication and severe disease [7,8]. Most strongly neutralizing human antibodies bind to quaternary structure epitopes that span two or more E proteins on the viral particle, whereas poorly neutralizing antibodies mainly bind to simple epitopes preserved within a domain of the E protein. While antibodies targeting quaternary structure epitope on E proteins are protective and block pH-induced fusion [9], the relationship between antibody cross-linking of E molecules and neutralization potency has not been experimentally tested. To better define the structural basis and mechanism of ZIKV neutralization by human antibodies, here we focused on human monoclonal antibody (mAb) G9E, which strongly neutralizes multiple ZIKV strains, but not dengue virus (DENV) and protects mice from a lethal ZIKV challenge [10]. Previous studies point to G9E binding to a unique (Zika type-specific) epitope on domain II of E protein that may partially overlap with a highly conserved epitope targeted by EDE1 human mAbs that neutralize DENVs and ZIKV [10–12]. Using X-ray crystallography and cryo-electron microscopy (cryo-EM), we mapped the binding site of G9E to a quaternary epitope that spans two E molecules forming a homo-dimer. Using the fine footprint of the G9E and structure-guided G9E paratope mutants, we evaluated the contributions of antibody binding affinity versus antibody-mediated E protein cross-linking to virus neutralization. Our results demonstrate that the ability of the antibody to cross-link E monomers is the main determinant of neutralization potency.

## Results

### G9E targets an immunodominant quaternary epitope on the ZIKV E-protein

To define the footprint of G9E bound to ZIKV E protein, we expressed and purified the ectodomain of ZIKV E and the antigen binding (Fab) fragment of G9E. The G9E Fab retained

neutralizing activity against ZIKV (strain H/PF/2013) in a cell culture focus reduction neutralization test (FRNT) (S1 Fig). The ZIKV-E/G9E-Fab complex was purified by size exclusion chromatography, and the crystal structure of the complex was determined by molecular replacement. The resulting electron density map resolved the polypeptide chain and revealed the fine details of E protein interaction with G9E at 3.4 Å resolution (Figs 1 and S2).

The crystallographic asymmetric unit contains a hetero-hexameric subunit formed by two G9E Fab fragments and one ZIKV E homodimer (Fig 1A). Both Fabs exhibited a similar mode of binding to one E homodimer. Structural alignment between the two copies of the E proteins or the two copies of the heavy or light chains revealed that they are highly similar, judged by the low root mean square deviations (0.34–0.53 Å for equivalent Ca atoms). While overall E protein structure was largely retained upon binding to G9E Fab, superimposition of E monomers in the G9E Fab complex and the previously determined soluble ZIKV E protein structure (PDB ID 5JHM) showed global domain shifts in EDII and EDIII. This included inward movements of the fusion loop region by 2 Å, and the movement of EDIII by 3 Å toward the highly ordered EDI N154 glycan loop, causing an increase in the E homodimer interface by 120 $Å^2$ (S3 Fig). The highly flexible EDI-EDII hinge region is largely unchanged between these two structures. Thus, the ZIKV E protein in complex with G9E closely resembles the flat conformation observed in other soluble E protein structures but differs from the curved conformation of the E proteins observed on the mature virion.

Next, we used cryo-EM to understand further the structural basis for G9E recognition of the intact virion. We incubated purified mature ZIKV strain H/PF/2013 with the G9E Fab fragment at Fab:E protein molar ratios of 1.2:1 and determined the cryo-EM map of G9E Fab: ZIKV to an overall resolution of 5.9 Å, as measured by the gold standard FSC curve cutoff at 0.143. This map showed clear borders and shapes corresponding to the G9E Fab and E protein structures, including the helical ridges of the E protein transmembrane region (Fig 2A and 2D). The map revealed 180 Fabs binding to one virus particle (Fig 2A and 2B). The G9E Fab bound to E proteins located on 3-, 5- and 2-fold axes in the icosahedral asymmetric unit, and the epitope is located mainly on EDII and the binding mode is consistent with the crystal structure (Fig 2B and 2C). Superimposition of E dimer in the G9E Fab complex determined by the cryo-EM and the crystal structure showed global domain shifts in EDII in line with the curved conformation observed on the mature virion. This included inward movements of the fusion loop region by 6 Å (Fig 2E).

The ZIKV E/G9E Fab complex structure by X-ray crystallography and cryo-EM revealed that each Fab fragment cross-linked an E homodimer by binding to a quaternary structure epitope spanning the homodimer (Figs 1B and 2B). The G9E Fab footprint covers a buried surface area (BSA) of 988 $Å^2$, of which 73% (709 $Å^2$) comprises the majority of EDII of one E protein (Fig 1B, Site 1). The remaining BSA is formed on the adjacent homodimer E protein involving the EDI N154 glycan loop and the EDI-EDII hinge region (Fig 1B, Site 2). In each Fab, the heavy chain variable domain binds to both site1 and 2 and contributes ~74% (731 $Å^2$) to the BSA. In comparison, the light chain variable domain binds only to site 1, contributing $\sim 26\%$ (257 $Å^2$) to the BSA.

The G9E footprint on ZIKV E-protein overlaps with the binding sites of a previously characterized ZIKV neutralizing human mAbs, ZIKV-117 [13], ZIKV-195 [14], and Z20 [15] (Fig 1C and 1D). The high G9E neutralization potency (11 ng/ml) is similar to ZIKV-117 (6 ng/mL), 15-fold greater than ZIKV-195 (77–600 ng/mL), and >60-fold better than the Z20 (370 ng/mL) [10,13]. Both ZIKV-117 and Z20 were shown to recognize quaternary epitopes on EDII covering BSA of 1132 and 797 $Å^2$, respectively. While G9E and Z20 binding sites are entirely comprised within the E-homodimer, the ZIKV-117 binding site includes both E homodimer and E dimer-dimer interface, covering three E proteins in the raft [13,14]. Our

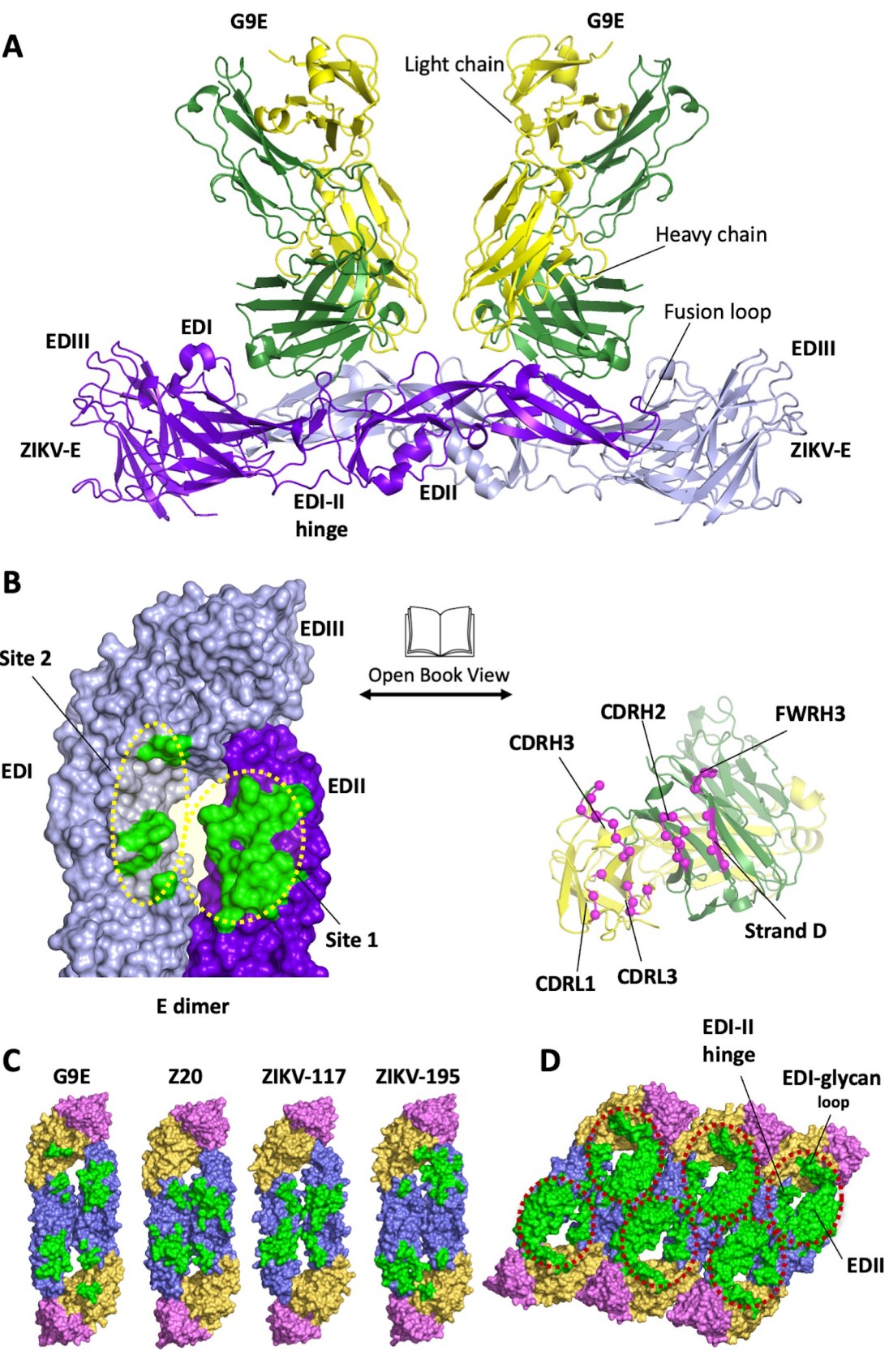

**Fig 1. G9E targets an immunodominant quaternary epitope on ZIKV E-protein.** (A) Structure of ZIKV E in complex with G9E. The structure reveals that two G9E Fab fragments bind the E-dimer in a similar mode. G9E Fab fragments and E-dimer are shown in cartoon representation (green—Fab heavy chain; gold—Fab light chain; ZIKV E dimer (protomer 1—purple; protomer 2—lavender). (B) G9E binds a quaternary epitope formed on E-dimer. Open book representation of the interface formed between E-dimer (green) and G9E (pink) are shown. The quaternary epitope comprises a major (site 1) and a minor (site 2) site on E-dimer. The paratope comprises heavy and light chain CDRs. (C) G9E targets an immunodominant epitope centered on EDII. G9E footprint on ZIKV E-dimer overlaps with previously described neutralizing human mAbs ZIKV-117, Z20, and ZIKV-195 isolated from patients infected with ZIKV. The quaternary epitopes targeted by the respective neutralizing mAbs are shown in green. E domains are shown in orange (EDI), blue (EDII) and pink (EDIII). ZIKV-195 epitope was obtained from ZIKV/ZIKV-195 structure (PDB ID: 6MID). ZIKV-117 footprint was derived from the mAb (ZV-67) fitted in the Cryo-EM map of ZIKV/ZIKV-117 structure (PDB ID: 5UHY). ZV20 epitope was obtained from ZIKV/Z20 structure (PDB ID: 5GZO). (D) Combined EDII targeting antibody epitope defines an immunodominant region on ZIKV E-raft. Combined epitope comprised of G9E, Z20, ZIKV-117, and ZIKV-195 is shown within the red circle, including EDII, EDI-II hinge, and the EDI glycan loop.

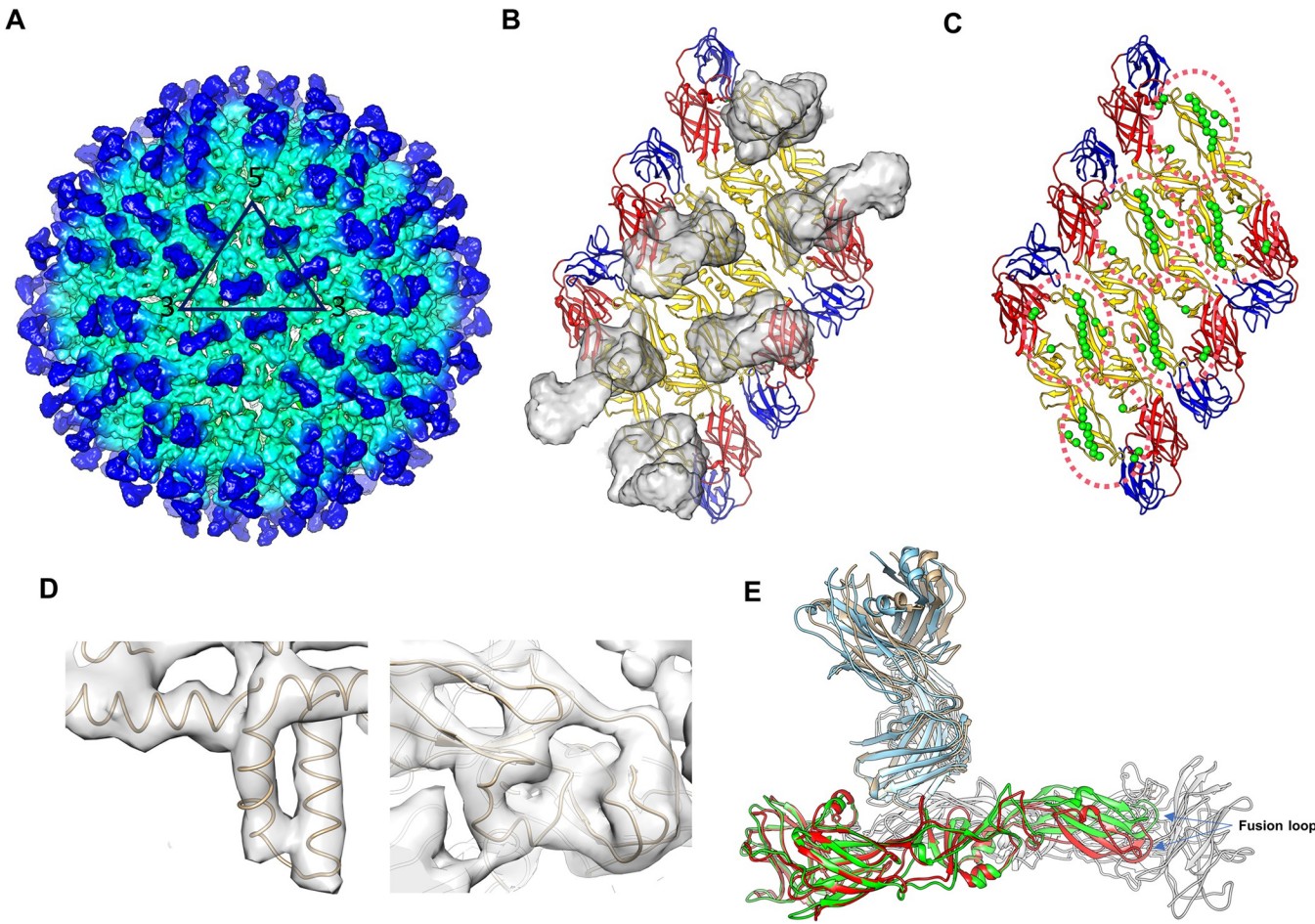

**Fig 2. Cryo-EM map of the ZIKV complexed with G9E Fab.** (A) The cryo-EM map of the ZIKV:G9E Fab complex was determined to 5.9 Å resolution. Densities corresponding to the E protein layer and Fabs are colored in cyan and blue, respectively. The black triangle indicates an asymmetric unit and the 5-, 3-, 2-fold vertices are labeled. (B) Densities of Fab molecules displayed on the three E proteins within a raft. The E protein EDI, EDII and EDIII are colored in red, yellow and blue, respectively. (C) Residues forming the G9E epitopes (pink dotted circles) are shown as green spheres. (D) Zoom-in views of the densities of the fitted trans-membrane α-helices (left) and EDII (right). (E) Comparison of the cryo-EM structure with the crystal structure of the soluble E:G9E Fab complex. When G9E Fab is bound to the virus surface, it did not induce structural changes to the E proteins (red). When superimposed onto this structure, the crystal structure shows its EDII domain (green) is elevated to represent the flat conformation of the soluble E protein.

cryo-EM map showed that all 180 binding sites on the surface of the virus were occupied by G9E Fab molecules, whereas ZIKV-117 occupied 60 sites (S1 Table).

## N67 glycan blocks binding of G9E and other ZIKV neutralizing antibodies

G9E binds to ZIKV but not to the four dengue virus (DENV) serotypes, which are closely related to ZIKV [10]. ZIKV E protein has a single N-linked glycan on EDI, whereas DENVs have two N-linked glycans, one on EDI and a second glycan at N67 on EDII. When comparing the sequence and structure of DENV and ZIKV E proteins (S4 Fig), we observed that the glycan at position N67 on DENVs partially masks the region on EDII that forms the G9E epitope. We predicted that introducing an N-linked glycan at position 67 on EDII of ZIKV would block the G9E epitope and prevent antibody binding (Fig 3A). To test this prediction, we generated ZIKV mutant rZIKV-D67N, which introduces an N-linked glycosylation site in ZIKV E analogous to the N67 site in DENV. The mutant virus was viable and confirmed by western blot to have an E protein of increased molecular weight consistent with an additional glycan (S5 Fig). Next, we compared the ability of different ZIKV-specific mAbs to bind and neutralize WT ZIKV and rZIKV-D67N. As predicted, G9E and Z20 were unable to bind or neutralize rZIKV-D67N (Fig 3B and 3C). Human mAbs that bind to EDIII (ZKA190 and Hu-ZV67), EDI (A9E), and the E dimer dependent epitope conserved between DENV and ZIKV (EDE C8) bound and neutralized both WT and rZIKV-D67N (Fig 3B and 3C). These results establish that a site on ZIKV EDII targeted by neutralizing human mAbs is blocked by adding a glycan at position N67, a site generally glycosylated in the four DENV serotypes but not other flaviviruses [16].

To evaluate if the ZIKV EDII neutralization site defined by mAb G9E was a target of serum neutralizing antibodies, we compared the ability of convalescent sera from ZIKV patients to neutralize WT ZIKV and rZIKV-D67N. The addition of the glycan at position 67 on EDII reduced the neutralization potency of 7 of 10 primary ZIKV immune sera tested (Fig 3D). Our results demonstrate that mAb G9E defines an antigenic region centered on EDII that is a major target of serum neutralizing antibodies in ZIKV patients.

## G9E neutralizes ZIKV before and after attachment to the cell surface

We next performed studies to define the mechanism of G9E mediated neutralization of ZIKV. In the mature virus, the E protein remains flat on the surface of the virion, burying the fusion loop within the head-to-tail E homodimer. Following viral attachment and entry into cells, the low pH environment within endosomes triggers the rearrangement of E proteins from homodimers to trimers leading to viral envelope fusion with endosomal membranes and the release of the viral nucleocapsid-RNA complex into the cytoplasm [17–20]. As G9E cross-links E molecules forming a single homodimer, we hypothesized that G9E neutralizes ZIKV by blocking conformational changes required for viral membrane fusion within endosomes.

If G9E mainly neutralizes ZIKV by blocking viral membrane fusion and not viral attachment to cells, the mAb should be able to neutralize ZIKV after the virus attaches to the cell surface (Fig 4A). Initially, we assessed the ability of ZIKV to bind to the cell surface in presence of G9E. We incubated ZIKV with increasing concentrations of G9E and then incubated the virus/mAb mixture with Vero-81 cells at 4°C to allow viral attachment to the surface but not entry. After washing the cells to remove any unbound virus, we measured levels of cell-associated ZIKV RNA by qRT-PCR. In this pre-attachment assay, G9E reduced the relative amount of ZIKV RNA associated with Vero-81 cells by approximately 50% at a concentration of 200 ng/mL (Fig 4B). We subsequently performed the same protocol but now placed virus/mAb/ cell mixture at 37°C to measure infection by FRNT. At 200ng/mL, G9E completely blocked the

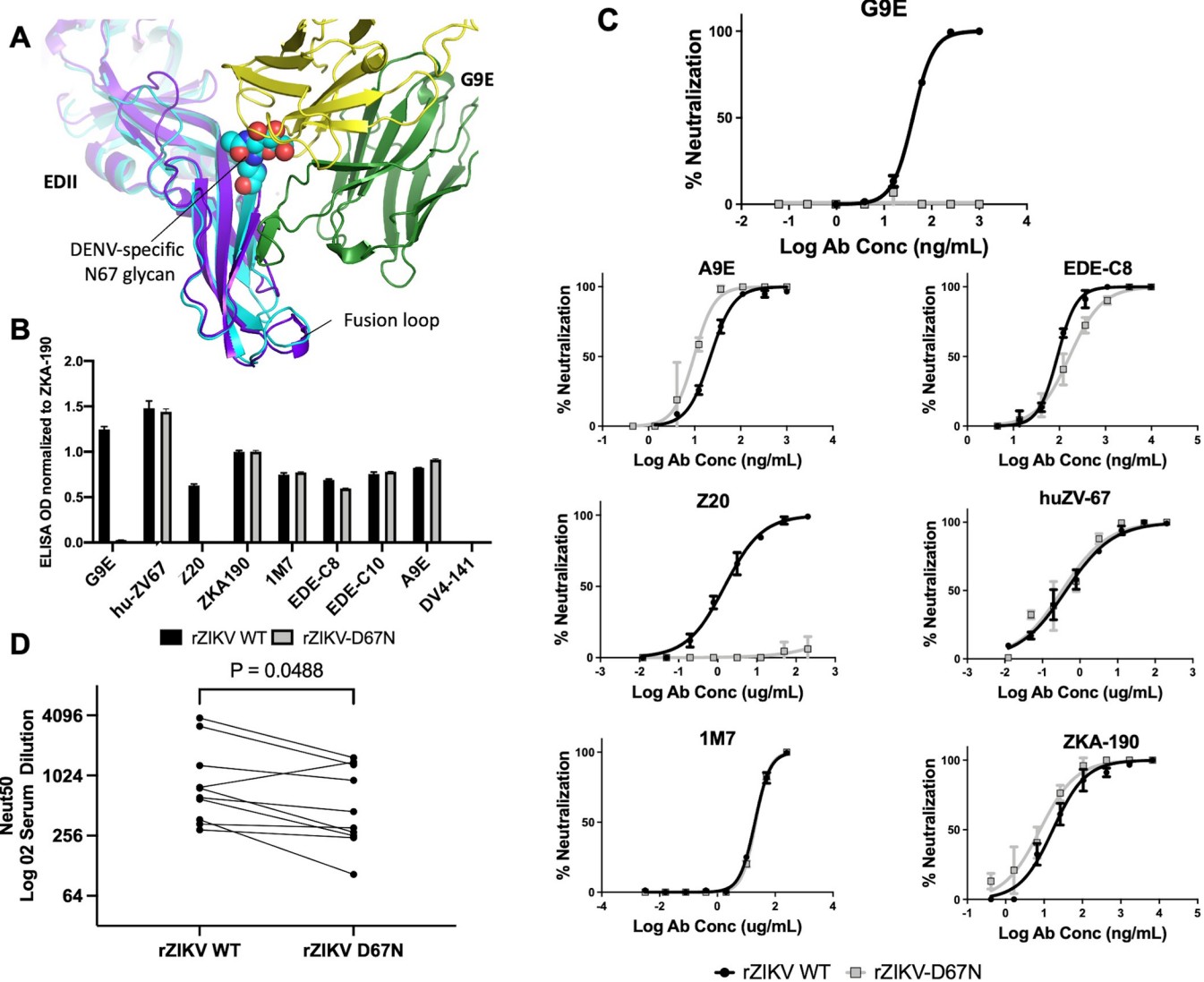

**Fig 3. An engineered N67 glycan on ZIKV E protein blocks G9E binding and neutralization.** (A) Structural overlay of ZIKV E/G9E complex on to DENV2 E-protein. Introducing a 'DENV-like' N-67 glycan on ZIKV E protein was predicted to block G9E binding. DENV E (cyan, PDB 1OAN), ZIKV E (purple), G9E heavy-chain (green), and G9E light-chain (gold) are shown in cartoon representation. DENV N-67 glycan is shown as spheres. (B) G9E does not bind to a recombinant ZIKV with N-linked glycan on EDII. Comparison of ZIKV mAbs binding to ZIKV H/PF/2013 and ZIKV H/PF/2013 with engineered glycan at position 67 (rZIKV-D67N). ZIKV type-specific mAbs ZV67 and ZKA-190 bind to EDIII. ZIKV type-specific mAbs G9E and Z20 target quaternary structure epitopes on EDII. MAbs EDE-C8 and EDE-C10 target quaternary structure E dimer dependent epitopes that are conserved between ZIKV and DENVs. mAb 1M7 binds to a highly conserved epitope near the fusion loop. DV4-141 is a DENV4 specific mAb used as a negative control. Antibody binding was evaluated on a ZIKV capture ELISA. Each mAb was tested in 3–4 independent experiments and the graph depicts representative data from one experiment. Within each experiment, the antibodies were tested in duplicate, and the binding signal was normalized to ZIKV EDIII targeting mAb ZKA-190. The graph displays the normalized mean OD and standard deviation (error bars). (C) Comparison of virus neutralization against ZIKV and rZIKV-D67N by a panel of ZIKV mAbs. While the integrity of the epitopes targeted by ZIKV mAbs is maintained, G9E failed to neutralize r-ZIKV-D67N. Each graph depicts representative results from 1 of two independent experiments, except for MAb 1M7 and EDE-C8, which were tested once. Within each experiment, the antibodies were tested at different concentrations in duplicate, and the graphs display mean % neutralization and standard deviation (error bars). (D) Convalescent sera from Zika patients show reduced neutralization against rZIKV-D67N. Convalescent sera from 10 individuals who experienced primary ZIKV were tested for neutralization WT ZIKV and rZIKV-D67N. Subjects 1–4 were tested in 2–3 independent experiments. Because of the small volumes of sera available, subjects 5–10 were tested just once. Paired neutralization titers against WT and D67N ZIKVs were compared using the Wilcoxon matched pairs test.

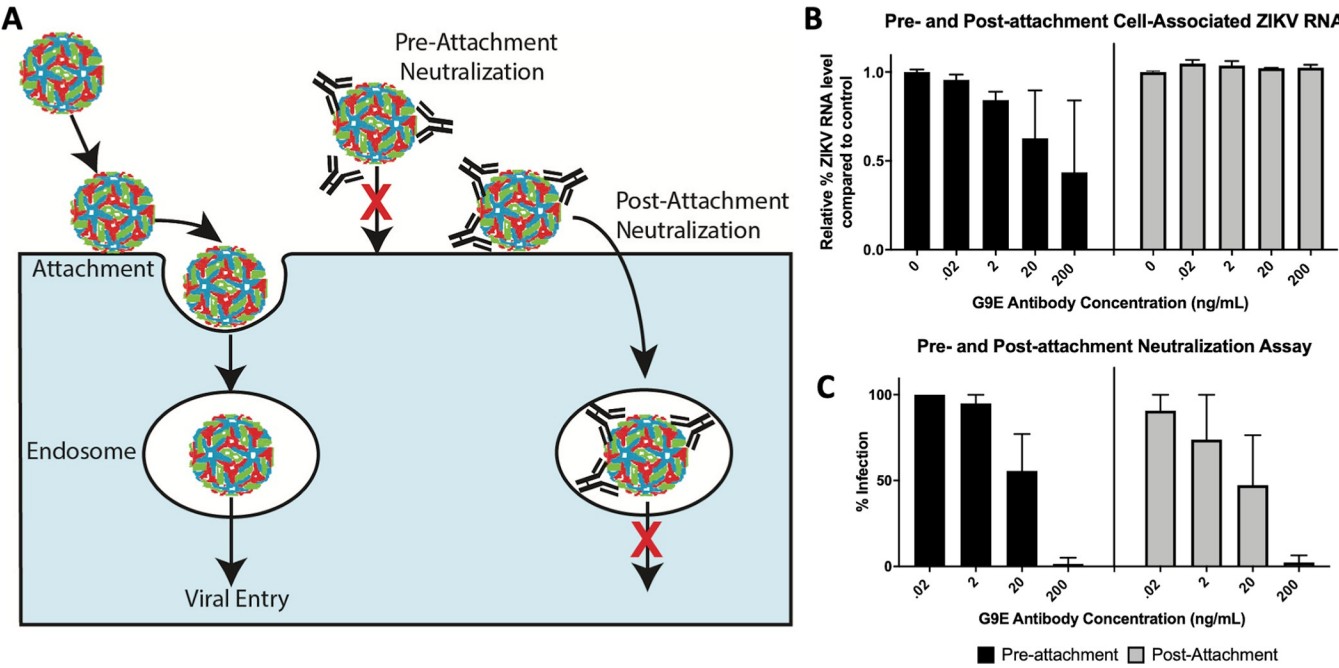

**Fig 4. MAb G9E blocks steps in entry after ZIKV attachment to cells. (A)** Schematic of ZIKV entry into the target cell. After attachment to the cell surface, the virus moves to endosomes, where a drop in pH triggers membrane fusion and viral RNA release to the cytoplasm. Neutralizing mAbs can block steps before (pre-attachment neutralization) or after (post-attachment neutralization) virus attachment to the cell surface. **(B and C)** The mechanism of G9E neutralization was assessed by adding the mAb before or after the virus attached to cells. In the pre-attachment neutralization assay, ZIKV and different quantities of G9E were incubated together before adding the virus to cells (black). In the post-attachment assay, ZIKV was allowed to attach to the cell surface before adding different quantities of mAb (grey). Cells were harvested to measure the initial amount of virus that bound to the cell surface in the presence of the antibody (B) and the number of cells that were productively infected 24hrs later (C). Two independent experiments were performed to determine levels of cell surface associated virus (B) and four independent experiments (C) were performed to determine the percentage of infected cells. Results from representative single experiments (mean of technical duplicates) are depicted in panels B and C. Error bars represent the standard deviation of the mean (SD).

ability of the virus to infect the cells (Fig 4C). These results indicate that G9E is able to block attachment and steps after attachment required for infection because the virions that bound to the virus in the presence of the mAb were unable to infect cells. To more directly assess if G9E was able to block infection after attachment, we preincubated ZIKV with Vero-81 cells at 4˚C. After washing the cells to remove any unbound virus, we added increasing quantities of mAb for 1 hr at 4˚C and then measured levels of cell-associated ZIKV RNA or ZIKV infection. Under these conditions, G9E was unable to displace virions already bound to cells but was able to block productive infection of cells at concentrations >20ng/mL (Fig 4B and 4C). These results demonstrate that G9E neutralizes ZIKV by blocking steps in viral entry before and after viral attachment to cells.

## Neutralization potency of G9E is dependent on binding across the E-dimer

Our hypothesis that G9E neutralizes ZIKV by cross-linking dimers and preventing fusion implies that a single Fab binding to both sites 1 and 2 is required for the potent neutralizing activity of the mAb. To gain deeper insights into specific G9E-ZIKV E interactions and their contribution to virus neutralization, we analyzed the paratope-epitope interface regions by PISA (Protein Interfaces, Surfaces, and Assemblies).

This analysis revealed that G9E binding to ZIKV E is predominately driven by hydrophilic and electrostatic interactions involving the heavy chain complementarity-determining regions (CDR) H3, CDR H2, and framework region of the heavy chain (FWR) H3 and the light chain

CDR L1 and CDR L3 with the lateral ridge region of EDII (site 1), the EDI 154-glycan loop and KL-hairpin and FG loop at the EDI-EDII hinge region (site 2) (Figs 5A and S4). Four regions on the heavy chain contact site 1 and site 2 residues on the E dimer. First, the CDR H3 (N107, W109, E111) connects to the exposed edge of the β-strand (D67, M68, S70, S72, and R73) of EDII by four backbone and two side-chain hydrogen bonds, thereby distinctively extending to the BDC β-sheet on the EDII lateral ridge. Second, the side chains of D53, D54, and S56 of CDR H2 form a salt bridge and hydrogen bonding interactions with R252 from the J strand of EDII. Notably, we previously reported that R252 is critical for G9E interaction by alanine scan screening analysis [10]. Third, CDR H2 mediates additional contacts involving the side-chains of D57, and Q58 with D278, and K209 from KL-hairpin and FG-loop, respectively at the EDI-EDII hinge region (site 2). Lastly, K76 from the FWR H3 forms an electrostatic interaction with E159 on the 154-glycan loop (Fig 5A and 5B). In comparison, the light chain contacts involve only site 1 mediated by four hydrogen-bonding interactions between G31, Y32, Y34, and Y93 (from CDR loops L1 and L2) and S66, D67, and K84 (from stands B and E of EDII) (Fig 5A).

To understand the significance of G9E cross-dimer interactions with site 2 for ZIKV neutralization, we generated three G9E Fabs introducing paratope mutants (S6 Fig). These mutants were designed to perturb the ZIKV E–G9E interactions in site 1 (G9E-S1), site 2 (G9E-S2) or site 1 and 2 (G9E-S1/2) (Figs 5 and S4). The G9E-S1 contained two alanine substitutions in the CDR H3 at position 109 and 111 (W109A and E111A). The W109A and E111A mutations were expected to disrupt the β-strand addition to the BDC β-sheet on the EDII lateral ridge. G9E-S2 contains three mutations (D57S, Q58A, K76S) that would eliminate the weak interactions between G9E and site 2. G9E-S1/2 (D53A, D54A, D57S, Q58A, K76S) has the same mutations as in G9E-S2 and additional mutations at position 53 and 54 were designed to reduce salt-bridge interaction with R252 in site 1.

Next, we tested the binding and neutralization activities of WT and mutant G9E Fabs with ZIKV. G9E-S1 (W109A, E111A) with disrupted EDII lateral ridge interaction showed >1500-fold reduced binding compared to WT G9E and entirely lost neutralization activity, demonstrating that site 1 vastly contributes to G9E binding to ZIKV. Without the significant site 1 interaction, site 2 completely lacks functional activity (Fig 5B and 5C). G9E-S2 with site 2 mutations showed ~10-fold reduction in binding, whereas its neutralization activity sharply decreased by >50-fold compared to WT G9E (Fig 5C and 5D). Adding mutations that reduce site 1 "R252" interaction onto site 2 mutant (G9E-S1/2) further weakened ZIKV binding by ~30-fold more than G9E-S2, though its neutralization activity was still comparable to G9E-S2 (Fig 5B and 5C). These data suggest that G9E interaction with E homodimer via site 2 plays a critical role in enhancing the potency of neutralizing activity, even though its isolated contribution is minimal to the overall binding of G9E to ZIKV E protein.

## G9E binding across E-dimer is necessary for blocking low pH-triggered viral fusion

We previously observed by cryo-EM that ZIKV virions aggregate when exposed to low pH (pH 5.0). This is due to the individual virus particles fusing with each other, probably the result of E proteins flipping up at low pH and attaching to membranes of adjacent viral particles. This process mimics the structural changes necessary during virus entry for low pH triggered fusion of ZIKV particles with endosomal membranes to release the viral genome into the cytoplasm of the target cell. We used cryo-EM to test if Fab G9E or its mutants could block pH-triggered fusion. Fab G9E or the mutants were initially allowed to bind to ZIKV at pH 8.0, mimicking the extracellular environment. We subsequently lowered the pH from 8.0 to either

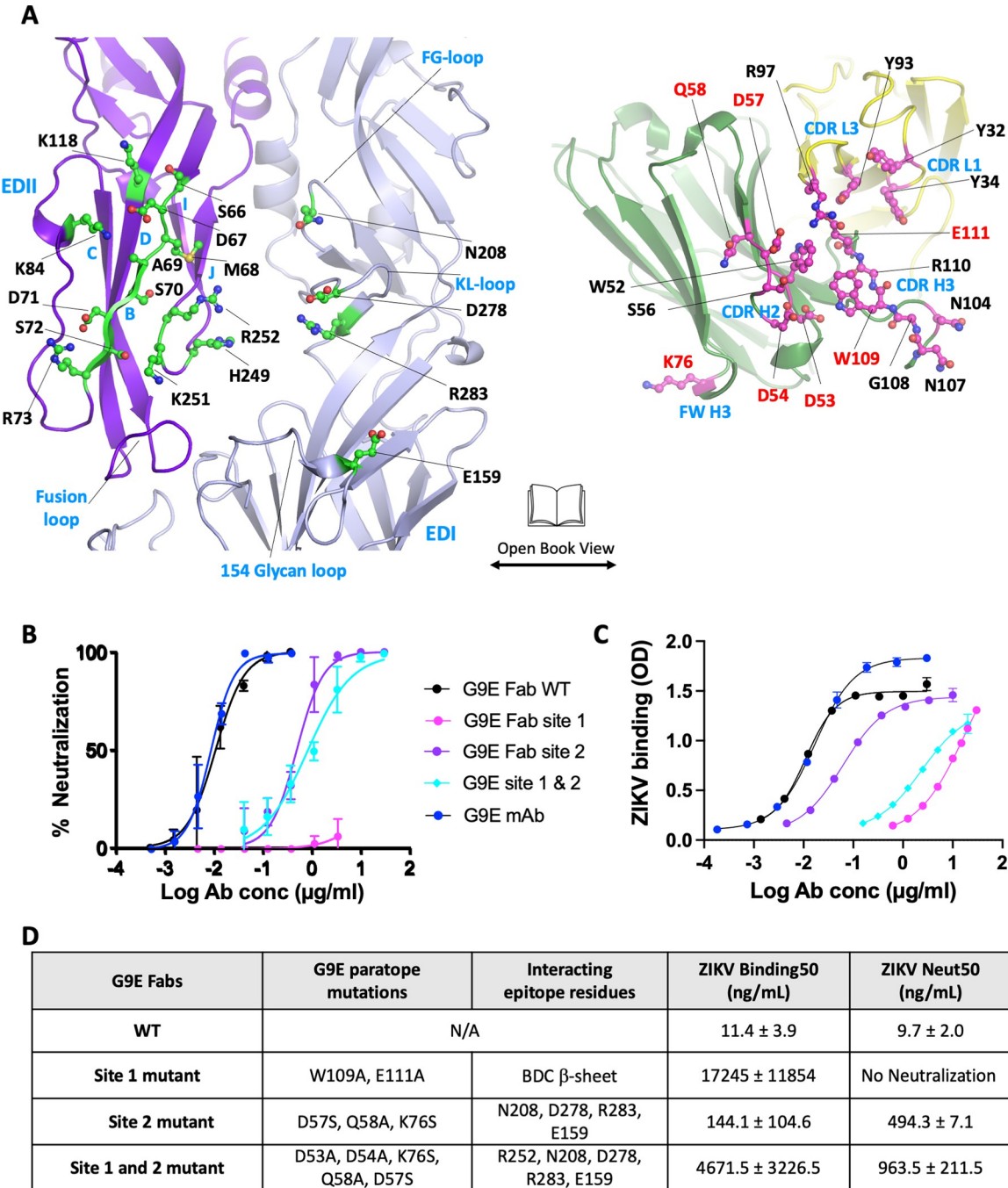

**Fig 5. G9E cross-dimer binding underlies potent neutralization of ZIKV.** (A) Close up view of the interactions between ZIKV E-dimer and G9E is shown in an "open book" representation. The interfacing residues between ZIKV E-dimer and G9E were identified by Protein Interfaces, Surfaces, and Assemblies program (PISA). Contact residues on E protein (green) and G9E (pink) are shown in ball and stick representations. The paratope residues selected for site-directed mutations are highlighted in red text. (B) Representative ZIKV neutralization curves for G9E WT and paratope mutant Fabs, G9E-S1, G9E-S2, or G9E-S1/2. (C) Representative ZIKV binding curves for G9E WT and paratope mutant Fabs, G9E-S1, or G9E-S2, or G9E-S1/2. Error bars (in panel B and C) are standard deviations of technical replicates. (D) Characteristics of ZIKV binding and ZIKV neutralization by G9E WT and paratope mutant Fabs. The neutralizing and binding data (in panel D) are the means and standard deviations of EC50 values obtained from three (binding) or two (neutralizing) independent experiments with technical replicates for each titration point.

6.5 (early endosome pH) or 5.0 (late endosome pH). The controls consisted of virus particles at the respective pH conditions without antibody (uncomplexed). Cryo-EM micrographs of uncomplexed ZIKV particles at pH 8.0 mainly showed particles with a smooth surface (Fig 6). At pH 6.5, most uncomplexed ZIKV particles were similar to those observed at pH 8.0, but some were distorted (Fig 6). At pH 5.0, the uncomplexed virus particles appeared spiky and formed aggregates, indicative of E protein fusion peptide exposure and insertion into adjacent viral particles. Fab G9E bound to the virus particles and prevented aggregation at pH5, demonstrating that Fab G9E inhibits fusion (Fig 6). G9E-S1 with disrupted EDII lateral ridge interaction in site 1 failed to bind to the virus at any pH and did not block low pH triggered fusion (Fig 6). Fab G9E-S1/2 with the mutations targeting "R252" interaction in site 1 and the entire site 2 interaction bound to ZIKV but failed to block virus aggregation at pH 5. G9E-S2 with the mutations abolishing binding to just the minor binding site (Site 2) on the adjacent E protomer, while preserving binding to the main site (Site 1) on EDII bound to ZIKV at pH 8 and 6.5 but failed to block the virus aggregation at pH 5. These observations suggest that the ability of G9E to cross-link E homodimer via site 2 is required for blocking low pH-triggered fusion of ZIKV particles, and site 1 binding alone lacked fusion inhibitory function.

## Discussion

While people infected with flaviviruses develop robust and long-lived memory B-cells (MBC) and circulating antigen-specific antibody responses, only a small fraction of these antibodies are responsible for functional virus neutralization and protection [21]. Recent studies have established that many potentially neutralizing and protective antibodies bind to quaternary structure epitopes that span two or more viral E proteins [13,21–24]. While the structure of human antibodies bound to flaviviruses or E protein complexes is consistent with antibody cross-linking of E proteins as a neutralization mechanism, direct experimental support for this has been lacking [22,24,25]. The nature of the ZIKV G9E footprint, consisting of a single dominant binding site on one E monomer and weaker peripheral contacts extending to the second E monomer, provided us a unique opportunity to test the significance of the minor intra-dimer contacts to virus neutralization. Our results strongly support that antibody mediated crosslinking of E proteins plays a critical role in the mechanism of virus neutralization. While we did not directly measure the impact of G9E binding on membrane fusion, our results establish that G9E inhibits low pH-induced changes to the virion, supporting a model in which E protein crosslinking is the major mechanism of neutralization. Our experimental data linking the G9E mediated E protein cross-linking to functional neutralization is likely to be broadly applicable to how other potent human antibodies block flavivirus infections.

Among the well-studied pathogenic flaviviruses, ZIKV is most closely related to the DENV complex. The antigenic region on EDII defined by G9E is likely the target of ZIKV-specific neutralizing and protective antibodies because the corresponding site on DENVs is masked by a N-linked glycan at position N67. While E glycosylation at position N154 is broadly conserved among flaviviruses, DENV is unique in having an additional N-linked glycan at N67. We predict that the region defined by G9E is likely to be a major target of neutralizing antibodies against other flaviviruses such as Japanese encephalitis, West Nile, and Spondweni viruses that do not belong to the DENV complex. While individuals exposed to sequential DENV serotypes often develop durable cross-neutralizing antibodies to all 4 serotypes, this response does not reliably extend to ZIKV [8,26,27]. Individuals sequentially infected with DENV followed by ZIKV also do not reliably develop durable DENV-ZIKV cross-neutralizing and cross-protective Ab responses [8,26]. Instead, these individuals develop distinct type-specific neutralizing Ab responses to the original DENV serotype and to ZIKV [26]. While not directly

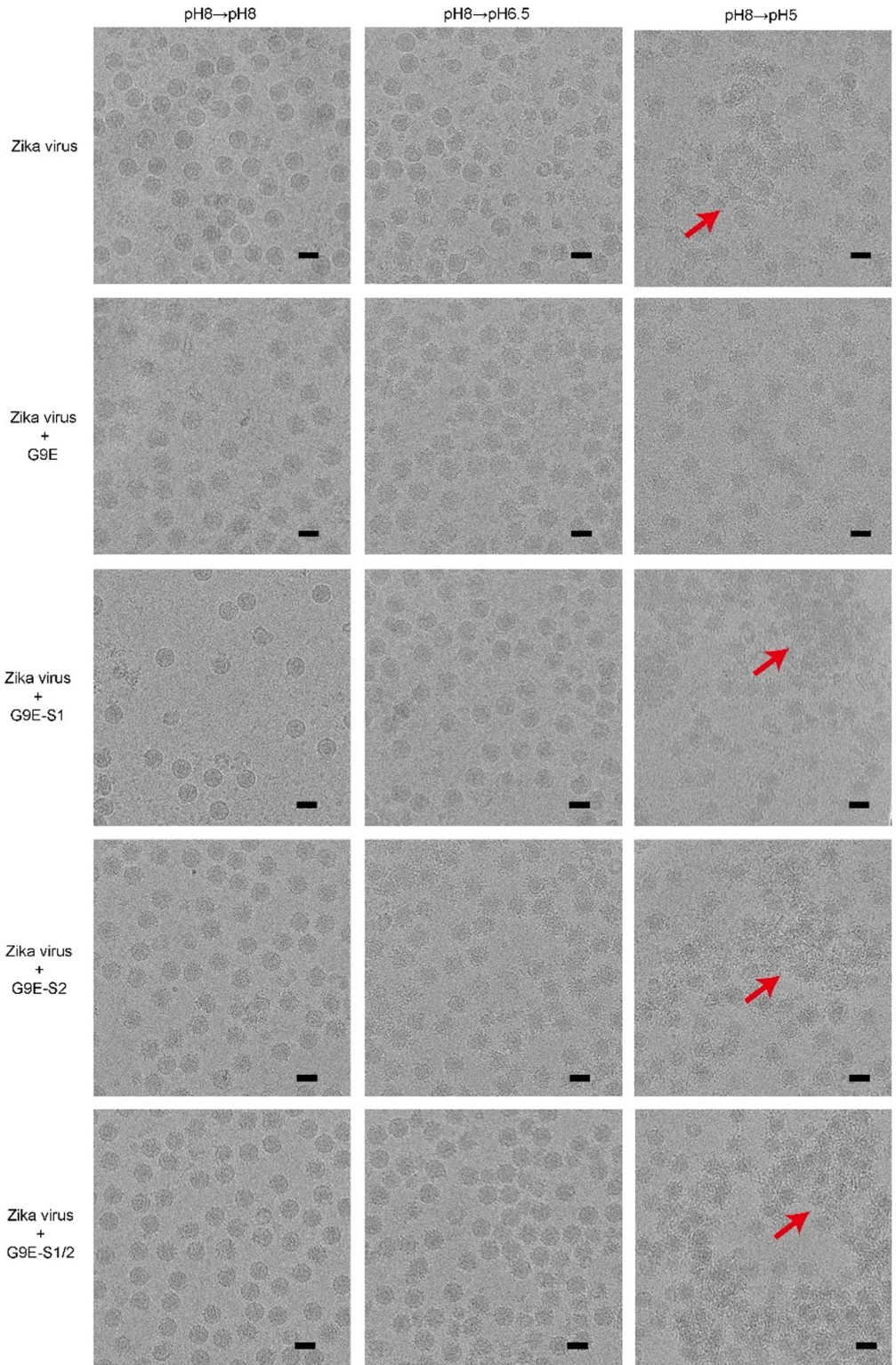

**Fig 6. Cryo-EM micrographs of the G9E Fab and its mutants complexed with Zika virus under various pH conditions: pH 8.0→pH 8.0 and pH 8.0→pH 6.5, and pH 8.0→pH 5.0.** In the uncomplexed ZIKV control, virus particles are largely smooth surfaced at pH 8.0. At pH 8.0→pH 5.0, the virus particles have a disordered surface and have aggregated (red arrow). When ZIKV is complexed with G9E Fab, the virus particles under all pH conditions appear spiky, indicating Fab binding. At pH 8.0→pH 5.0, no aggregation is detected, which suggests G9E Fab can

inhibit virus-virus fusion. When ZIKV is mixed with Fab G9E-S1, at pH 8.0→pH 8.0, virus particles remain smooth surfaced, similar to that in the uncomplexed ZIKV control, indicating Fab G9E-S1 could not bind to ZIKV. When ZIKV is complexed with either G9E-S2 or G9E-S1/2, at pH 8.0→pH 8.0 and pH 8.0→pH 6.5, the particles appear spiky, indicating that Fabs bind them. At pH 8.0→pH 5.0, aggregation is observed (red arrow), suggesting that these Fabs cannot inhibit virus-virus fusion. Scale bars, 500 Å.

addressed here, the presence of an N-linked glycan at position N67 in the 4 DENV serotypes but not on ZIKV may be key to understanding antibody neutralization patterns between DENVs and ZIKV. Recently, individuals immunized against Yellow Fever virus have been observed to develop neutralizing antibodies that are less effective against South American strains with a N-linked Glycan at position 269 compared to African strains that are not glycosylated [28].

In conclusion, a hallmark of flavivirus infections is the induction of rare but potent, quaternary epitope directed neutralizing antibodies that are correlated with long-term protection. We have identified an antigenic region on ZIKV that is a major target of type-specific neutralizing antibodies in individuals exposed to ZIKV infections. Our findings highlight the importance of antibody-mediated cross-linking of E proteins in the pre-fusion conformation as a mechanism for neutralizing flaviviruses.

## Materials and methods

### Expression and purification of recombinant ZIKV E proteins and antibodies

The G9E sequence was obtained during initial characterization of this mAb [10]. Z-20 and ZKA190 mAb sequences were obtained from PDB entries 5GZO, and 5Y0A respectively. A humanized version of ZIKV specific mouse EDIII mAb (HuZV-67) in IgG1 format was generated using the sequence obtained from PDB entry 5KVG. A codon optimized synthetic gene encoding for WT or mutant heavy or light chain mAb or Fab was cloned into a mammalian expression plasmid pAH. A human serum albumin secretion signal sequence was included at the 5'-end of each construct to enable secretion into the culture medium. The Fab heavy chain constructs also contained a 6xHistidine tag at the 3'-end. Recombinant Fab or mAb was expressed in Expi293 mammalian expression by co-transfection of heavy and light chain plasmids at 1:1 ratio. Recombinant Fab proteins were purified from the culture supernatant by nickel-nitrilotriacetic acid agarose (Qiagen). Recombinant mAbs were affinity purified by MabSelect resin (Cytiva, #17543802). Recombinant ZIKV E-protein and a cysteine crosslinked stable ZIKV E-protein dimer (A264C) with C-terminal 6x His-tag were expressed in the Expi293 cells and purified as described before [29]. Anti-flavivirus MAbs 2H2 (ATCC HB-114) and 4G2 (ATCC HB-112) were produced in hybridoma cell line by the UNC Protein Expression and Purification Core Facility. Purified protein products were verified by SDS-PAGE reducing gel.

### ZIKV/G9E Fab complex crystallization and structure determination

ZIKV E/G9E Fab complex was formed by mixing purified recombinant ZIKV E-protein and G9E Fab in solution at 1:1.2 ratio at room temperature for 30 min. The ZIKV E/G9E Fab complex was purified by Superdex 200-increase size exclusion chromatography column. Crystallization screening and optimization of ZIKV E/G9E Fab complex were performed in mosquito robots at the UNC's Center for Integrative Chemical Biology and Drug Discovery using the sitting-drop vapor-diffusion method. Crystals of ZIKV E/G9E Fab complex were grown by mixing 150 μl protein solution at 2.5 mg mL−1 and 150 μl crystallant solution consisting of 100

mM HEPES pH 7.5, 10%(w/v) PEG 8000. X-ray diffraction data were recorded on a MAR-225 CCD detector at the APS SER-CAT 22-BM beamline. Reflections were processed and scaled in HKL2000. Phases were obtained by molecular replacement using the structures of ZIKV E protein (PDB ID: 5JHM) and Fab (PDB ID: 4NKI) as templates. An initial search using the complete PDB coordinates of ZIKV E protein or Fab as a model was unsuccessful. Instead, four fragments of template structures encompassing ZIKV EDI-EDII (1–301 aa), ZIKV EDIII (302–406 aa), heavy and light chains of Fab molecules were used to phase the structure of ZIKV E/G9E Fab complex using Phaser. The translation function Z-scores (TFZ) were >8 for 7 of 8 solutions. The log-likelihood gain increased from 116 (1st solution) to 1444 (8th solution) as each component of the solution was added. Iterative refinement and model building were performed using PHENIX and Coot, respectively. The initial R-factor for the MR solution was 44%. After 3 cycles with a combination of rigid body, XYZ, and group-B refinement strategy, the starting R-factor and R-free were 29.9% and 32.3%, respectively. The final R-factor and R-free reported in the S2 Table are 23.2 and 25.9%, respectively. Our early refinement strategy included torsion angle simulated annealing to eliminate model bias. Electron density maps for the regions, for example, the E 154 glycan loop and the CDR loops, which were absent in the template model, became interpretable. We used Group-B factors and torsional angle noncrystallographic symmetry (NCS) restraints throughout the refinement cycles, and the final refinement cycles had TLS refinement utilizing a total of 6 TLS groups (1 TLS group per chain). The temperature factors for both heavy and light chain CDR loops are low, and the electron density corresponding to these regions are interpretable to confidently assign main and side-chain atoms. A representative electron density map, B-factor distribution and Molprobity multicriterion-plot for CDRs are shown in S2 Fig. The data collection and refinement statistics are given in 2. The refined model of ZIKV E/G9E Fab complex had six protomers in the asymmetric unit. Molecular figures were generated in PyMOL and interaction analysis were performed in PISA.

## Virus sample preparation for cryo-EM studies

*Aedes albopictus* C6/36 cells were grown in RPMI 1640 media supplemented with 10% fetal bovine serum at 29˚C. At about 80% confluency, the cells were inoculated with ZIKV strain H/PF/2013 at a multiplicity of infection of 0.5 and incubated at 29˚C for 3 days. Tissue culture supernatant was clarified by centrifugation at 9000 g for 1 h. Virus was precipitated overnight from the supernatant using 8% (w/v) polyethylene glycol 8000 in NTE buffer (12 mM Tris-HCl pH8.0, 120 mM NaCl and 1 mM EDTA) and the suspension was centrifuged at 14,400g for 1 h. The pellet was resuspended in NTE buffer and then purified through a 24% (w/v) sucrose cushion followed by a linear 10–30% w/v potassium tartrate gradient. The virus band was extracted, buffer exchanged into NTE buffer and concentrated using a concentrator with 100-kDa molecular weight cut-off filter. All steps of the purification procedure were done at 4˚C.

## Cryo-EM sample preparation and Cryoelectron microscopy

For the cryo-EM reconstruction, the Fab G9E was mixed with ZIKV at a molar ratio of 1.2 Fab to every E protein. The mixture was incubated for 30 min at 4˚C followed by ~1 h on ice, and then the sample was collected on a Titan Krios (FEI) microscope equipped with 300 kV field emission gun. Leginon [30] was used for automated data collection. The calibrated magnification was 47,000, giving a pixel size of 1.71 Å. The images were recorded in single image mode on Falcon II direct electron detector (FEI) with a total dose of 20 $e^-Å^{-2}$. The images were taken at underfocus range between 1.0 and 3.5 μm. A total of 2140 micrographs were collected

for the complex. The astigmatic defocus parameters were estimated using Gctf. [31]. Particles were picked using Gautomatch, and subsequently subjected to Relion [32] to produce 2D class averages. Classes containing junk and broken particles were excluded from further processing. 4,465 particles in the Fab G9E:ZIKV complex samples were selected for further processing. The 3D refinement produced structures with resolution of 5.9 Å as measured by the gold standard Fourier shell correlation (FSC) cut-off of 0.143. Cryo-EM data collection, refinement and validation statistics are given in S3 Table.

For observation of the ability of Fabs to inhibit virus-to-virus fusion using cryo-EM, the Fab G9E and mutants were mixed with ZIKV at a molar ratio of 1.2 Fab to every E protein, respectively. The mixture was incubated for 30 min at 4°C followed by ~1 h on ice, and then applied to a Lacey Carbon grid (TED PELLA, INC) for 10 s prior to adjusting the pH. The final pH of the virus was reached by addition of a volume ratio of 1.8 μL of 50 mM MES buffer at respective pH (pH 5.0 or pH 6.5) to every 1.2 μL of the virus-Fab mixture. The pH-adjusted samples were left on the grid for another 15 s. The grid was then blotted with filter paper and flash frozen in liquid ethane by using the Vitrobot Mark IV plunger (FEI). ZIKV without Fab for each pH were prepared similarly as the controls. The images of the frozen ZIKV complexes were taken with the Titan Krios transmission electron microscope, equipped with 300 kV field emission gun, at nominal magnification of 47,000 for all the complex samples. A 4096 * 4096 FEI Falcon II direct electron detector was used to record the images.

## ZIKV infectious clone mutagenesis

We used a previously described infectious clone of ZIKV strain H/PF/2013 [33, 34]. Site-directed mutagenesis was used to introduce a glycosylation motif (N-X-S/T) at position 67 of the envelope protein (GAC ATG GCT > AAC ACG ACA). The resulting purified plasmids were digested (New England BioLabs), ligated, *in vitro* transcribed (mMachine T7 Ultra transcription kit from Ambion), and electroporated into Vero-81 cells as previously described [35]. Supernatants from electroporated Vero-81 cells were harvested after 6 to 7 days and passaged once on Vero-81 cells to generate virus stocks. Virus stocks were titered by FFA on Vero-81 cells. Envelope protein glycosylation status was confirmed by size shifts on western blots as previously described [34].

## Pre and Post-attachment assay

Pre and post-attachment assays were done as previously described [36]. Briefly, pre-attachment conditions added varying concentrations of G9E mAb to 60–80 foci of H/PF/2013 ZIKV and incubated 1 hour at 4°C. The virus mixed with antibody solution was added to the confluent layer of Vero-81 cells and incubated for 1 hour at 4°C. Post-attachment conditions added 60–80 foci of H/PF/2013 ZIKV to the confluent layer of Vero-81 cells for 1 hour at 4°C. Cells were washed of excess ZIKV with ice-cold DMEM/F12 media supplemented with 20 mM HEPES buffer. Varying concentrations of mAb were added to Vero-81 surface-bound ZIKV at 4°C. For both conditions, cell-associated viral RNA was harvested by adding trizol directly to the confluent cell layer and purifying RNA through QiaAMP viral mini kit. RNA was converted to cDNA by iScript Reverse Transcriptase Supermix (Biorad, #1708841) and detected using Sybr Green (Thermo, 4309155) system with primers specific for ZIKV E-protein (F: CCGCTGCCCAACACAAG, R: CCACTAACGTTCTTTTGCAGACAT) adapted from a previous publication [37]. In separate plates, the focus-forming assay was proceeded by heating the attached complex to 37°C and harvesting after 40 hours. Foci were detected by immunostaining with pan-Flavi antibody 4G2.

## ZIKV capture ELISA

96 well-high-binding titer plate (Greiner, 655061) was coated at 100ng/well with pan-flavivirus fusion loop mAb 4G2 in 0.1M carb buffer. The wells were blocked with PBS containing 3% skim milk and 0.05% Tween-20 for 1 hour at 37˚C. After washing, ZIKV in culture supernatant was added to each well and incubated for 1 hour at 37˚C for capture by 4G2. Serially diluted serum/mAb/Fab in blocking solution was added to the well and incubated for 1 hour at 37˚C. Plates were washed with TBS containing 0.2% Tween and incubated for 1 hour at 37˚C with Goat anti-human Fc (Sigma, A9544) or Fab (Jackson immunoResearch AB_2337617) specific IgG conjugated to alkaline phosphatase. Plates were washed, developed with p-Nitrophenyl phosphate substrate (Sigma, N1891), and absorbance was measured at 405 nm.

## Focus Reduction Neutralization Test (FRNT)

ZIKV FRNT assay was performed as described [38]. Briefly, mAbs were serially diluted in DMEM (Life Technologies, 11330032) media supplemented with 2% Fetal Bovine Serum (Sigma, TMS-013-B), 1% L-glutamine (Life Technologies 25030081), 1% penstrep (Mediatech, 30002CI), and 1% sodium bicarbonate (Life Technologies, 25080094) and incubated with H/PF/2013 ZIKV for 1 hour at 37˚C. Antibody and virus mixture was added to the confluent layer of Vero-81 cells in 96-well flat-bottom plate and incubated for 1 hour at 37˚C. Excess mAb/virus mixture was flicked off the plate, and 180 μL of Optimum (Life technologies, 31985070) supplemented with 2% methylcellulose was added to individual wells. The plate was fixed with 4% PFA and stained for ZIKV foci with flavivirus specific mAb after 40-hour incubation.

## Supporting information

**S1 Fig. Neutralization of ZIKV H/PF/2013 by G9E mAb and G9E Fab.** Representative ZIKV neutralization curves for G9E mAb (A) and G9E Fab (B), along with previously characterized ZIKV specific (C, B11F) and DENV2 specific (D, 2D22) mAbs, are shown. G9E mAb and G9E Fab were expressed in Expi293 mammalian cells and purified from the cell culture medium by MabSelect resin or nickel-nitrilotriacetic acid agarose resin, respectively. G9E Fab retains neutralization activity against ZIKV, similar to the parent G9E mAb. Error bars for each data point are standard deviations of the technical replicates. (E) EC50 values obtained from non-linear sigmoidal 4PL curve fitting are shown.
(TIF)

**S2 Fig. Correlation of map quality and B-factor in ZIKV-E/G9E Fab complex structure.** (A) Representative electron density map of G9E Fab/ZIKV E complex. An initial 2Fo-Fc electron density map (contour 1.0 sigma) of the ZIKV E/G9E complex illustrates that the starting phases obtained by molecular replacement were of excellent quality to reveal the nature of the interaction between G9E and ZIKV-E-protein. ZIKV E-protein (green) G9E Fab (yellow) are shown as sticks. (B) Thermal parameter distribution in ZIKV E/G9E Fab shown as B-factor "putty". The isotropic B-factors are depicted on the structure as spectrum range from 28.9 $\text{Å}^2$ (blue, lowest B-factor) to 238.7 $\text{Å}^2$ (red, highest B-factor), with the ribbon radius increasing from low to high B-factor. The mean B-factor was 120.67 $\text{Å}^2$. The lowest B-value was observed in the interfacing region between DII of E protein and the CDR regions of G9E Fab, where the electron density is well resolved. (C) Molprobity multicriterion-plot for CDRs. The likelihood-weighted 2mFo-DFc map and the Fc map calculated from the model were compared and real-space correlation coefficient for each residue were obtained. Comparison of the 2mFo-DFc

map, the Fc map, the real-space CC and the B-factor for each of reside in CDR loops are shown.
(TIF)

**S3 Fig. G9E binding induces a small domain motion in ZIKV E-dimer.** Structural superposition of the E-protein conformation of the template structure used for molecular replacement (PDB ID: 5JHM, yellow) and the E-protein conformation observed in complex with G9E (protomer 1—blue; protomer 2—purple). G9E induces a 2 Å inward movement of the fusion loop (blue strand, notated by red arrows) towards the EDI glycan loop (purple spheres) of the neighboring E-protein. G9E also causes a 3 Å inward movement of EDIII (purple strand, notated by the red arrow) towards its EDI glycan loop (purple spheres). These movements cause an increase in the E-dimer interface.
(TIF)

**S4 Fig. Details of the interaction interface. A.** Summary of interaction distance between E and G9E residues in the crystal structure of ZIKV E/G9E complex. A donor-acceptor atom distance of 4 Å between E and one of the Fab was considered as a H-bond. A distance of 6 Å was considered for a salt bridge. **B.** Close-up view of the interacting residues in E protein (purple) and G9E heavy (green) and light chains (yellow). The color of the residue name and number is matched to the carbon skeleton of amino acid. The main chain hydrogen bonding interaction between E and G9E are shown as dotted green lines. The heavy chain paratope residues selected for site-directed mutations are highlighted in red text. **C.** Amino acid conservation analysis within the G9E binding site. G9E interacting residues in ZIKV E protein was compared to the four serotypes of the DENV E proteins. ZIKV E protein residue number and name in single letter code are provided on the top rows. Based on the amino acid properties, DENV residues are categorized from favorable (green) to unfavorable (red). Identical residues are colored in grey. ZIKV E residue involved in main chain hydrogen bonding interaction is shown in cyan. DENV sequences were retrieved from NCBI using the accession codes provided within the parentheses: DENV1 (P17763), DENV 2 (GU289914), DENV3 (AAB69126), and DENV4 (AGS14893). The N-linked glycosylation at position 67 was predicted to sterically block G9E binding.
(TIF)

**S5 Fig. Western blot of rZIKV and rZIKV-D67N.** WT rZIKV retains a glycosylation site at position 154 on the envelope protein while DENV3 has two glycosylation sites at positions 67 and 154. rZIKV-D67N was created by introducing a glycosylation motif (DMA to NTT) at position 67–69. To indirectly assess glycosylation status, E-proteins were immunoprecipitated from Vero cells infected with WT rZIKV, rZIKV-D67N, or DENV3 and detected by western blot using flavivirus mAb 4G2 as primary antibody followed by HRP-conjugated goat anti-mouse IgG as secondary antibody. rZIKV-D67N exhibited a higher molecular weight E protein compared to WT rZIKV, consistent with the presence of an additional N-linked glycan.
(TIF)

**S6 Fig. SDS-PAGE analysis of the purified G9E Fabs.** G9E WT and paratope mutant Fabs were expressed in Expi293 mammalian cells and purified by Ni-NTA resin. Coomassie-stained SDS-PAGE run under reduced and non-reduced condition show the band corresponding to intact Fab and reduced Fab fragments.
(TIF)

**S1 Table. Characteristics of human ZIKV mAb targeting EDII.** Z20 [15], ZIKV-117 [13], and ZIKV-195 [14] are previously reported human mAbs isolated from patients who experienced ZIKV infection. G9E, ZIKV-117, and ZIKV-195 protected against ZIKV infection in a

murine model. The estimated binding sites on the ZIKV virion and the virus neutralization titer are tabulated.
(XLSX)

**S2 Table. Data collection and refinement statistics of ZIKV E/G9E complex structure.** Statistics for the highest-resolution shell are shown in parentheses.
(XLSX)

**S3 Table. Cryo-EM data collection, refinement and validation statistics.**
(XLSX)

## Acknowledgments

We acknowledge the use of the UNC's macromolecular crystallography core facility and 22-BM beamline (SER-CAT) at the Advanced Photon Source. We thank the support teams at both facilities for expert assistance, and we are grateful to Dr. Kenneth Pearce for his support with the use of Mosquito Robot for protein crystallization in screening and optimization stages. We thank Xinni Lim and Valerie S-Y Chew for virus purification, Shuijun Zhang, Victor Kostyuchenko and Guntur Fibriansah for technical support.

## Author Contributions

**Conceptualization:** Cameron Adams, Aravinda M. de Silva, Lakshmanane Premkumar.

**Data curation:** Cameron Adams, Derek L. Carbaugh, Bo Shu, Thiam-Seng Ng, Izabella N. Castillo, Ryan Bhowmik, Bruno Segovia-Chumbez, Ana C. Puhl, Stephen Graham, Shee-mei Lok, Aravinda M. de Silva, Lakshmanane Premkumar.

**Formal analysis:** Cameron Adams, Bo Shu, Shee-mei Lok, Aravinda M. de Silva, Lakshmanane Premkumar.

**Funding acquisition:** Sean A. Diehl, Helen M. Lazear, Shee-mei Lok, Aravinda M. de Silva.

**Investigation:** Cameron Adams, Derek L. Carbaugh, Bo Shu, Thiam-Seng Ng, Izabella N. Castillo, Ryan Bhowmik, Bruno Segovia-Chumbez, Ana C. Puhl, Stephen Graham, Sean A. Diehl, Helen M. Lazear, Shee-mei Lok, Aravinda M. de Silva, Lakshmanane Premkumar.

**Methodology:** Cameron Adams, Derek L. Carbaugh, Bo Shu, Thiam-Seng Ng, Izabella N. Castillo, Ryan Bhowmik, Bruno Segovia-Chumbez, Ana C. Puhl, Stephen Graham, Lakshmanane Premkumar.

**Project administration:** Sean A. Diehl, Helen M. Lazear, Shee-mei Lok, Aravinda M. de Silva, Lakshmanane Premkumar.

**Resources:** Sean A. Diehl, Helen M. Lazear.

**Supervision:** Sean A. Diehl, Helen M. Lazear, Shee-mei Lok, Aravinda M. de Silva, Lakshmanane Premkumar.

**Writing – original draft:** Cameron Adams, Bo Shu, Aravinda M. de Silva, Lakshmanane Premkumar.

**Writing – review & editing:** Cameron Adams, Derek L. Carbaugh, Bo Shu, Thiam-Seng Ng, Izabella N. Castillo, Ryan Bhowmik, Bruno Segovia-Chumbez, Ana C. Puhl, Stephen Graham, Sean A. Diehl, Helen M. Lazear, Shee-mei Lok, Aravinda M. de Silva, Lakshmanane Premkumar.

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
