## [Decision Letter · Decision Letter 0]

17 Oct 2022

Dear Dr. de Silva,

Thank you very much for submitting your manuscript "Structure and Neutralization Mechanism of a Human Antibody Targeting a Complex Epitope on Zika Virus" for consideration at PLOS Pathogens. As with all papers reviewed by the journal, your manuscript was reviewed by members of the editorial board and by several independent reviewers. The reviewers appreciated the attention to an important topic. Based on the reviews, we are likely to accept this manuscript for publication, providing that you modify the manuscript according to the review recommendations.

One reviewer has provided some additional queries that should be addressed related to figures with neutralization data. Additional details and clarity are required and should not require extensive revision.

Sincerely,

Richard J. Kuhn, PhD

Associate Editor

PLOS Pathogens

Ana Fernandez-Sesma

Section Editor

PLOS Pathogens

Kasturi Haldar

Editor-in-Chief

PLOS Pathogens

orcid.org/0000-0001-5065-158X

Michael Malim

Editor-in-Chief

PLOS Pathogens

orcid.org/0000-0002-7699-2064

One reviewer has provided some additional queries that should be addressed related to figures with neutralization data. Additional details and clarity are required and should not require extensive revision.

Reviewer Comments (if any, and for reference):

Reviewer's Responses to Questions

**Part I - Summary**

Reviewer #1: The added cryo-EM experiments at neutral and low pH greatly strengthen the study by providing evidence of Fab crosslinking to inhibit conformational changes.

Reviewer #2: The authors have addressed reviewer's concerns.

**Part II – Major Issues: Key Experiments Required for Acceptance**

Reviewer #1: (No Response)

Reviewer #2: (No Response)

**Part III – Minor Issues: Editorial and Data Presentation Modifications**

Reviewer #1: Figures with neutralization/ binding data still need clarification regarding independent experiments vs technical replicates, etc. Specifically:

Figure 3: The legend states that the data in panels B-D represents the average of two measurements. This does not specify whether these were independent experiments or technical replicates in one experiment. Error bars are not addressed. The data in Figure 3D is the basis for the following statement on lines 163-164: “Our results demonstrate that mAb G9E defines an antigenic region centered on EDII that is a major target of serum neutralizing antibodies in ZIKV patients.” While this may be true, it is too strong of a conclusion based on the information provided. Was only a single neutralization assay performed with technical replicates, or two independent experiments? Were the samples tested pairwise? There are no error bars.

Figure 4B and C: Again, are the measurements from a single experiment or two independent experiments?

Figure 5: The differences in neutralization curves would be easier to see if they were on the same graph, and additionally graphed as EC50 values. Still unclear on what is shown in panels B and C regarding # experiments and error bars. Are the neutralization curves a representative experiment of the three experiments used to calculate the average EC50 values in D? Same question with the binding data.

Supplementary Figure 1: What are the EC50 values? This could be clearly depicted by plotting EC50 values on a separate graph.

Reviewer #2: (No Response)

PLOS authors have the option to publish the peer review history of their article (what does this mean?). If published, this will include your full peer review and any attached files.

Reviewer #1: No

Reviewer #2: No

Figure Files:

Data Requirements:

Reproducibility:

References:

---

## [Editor Report · Decision Letter 1]

5 Dec 2022

Dear Dr. de Silva,

We are pleased to inform you that your manuscript 'Structure and Neutralization Mechanism of a Human Antibody Targeting a Complex Epitope on Zika Virus' has been provisionally accepted for publication in PLOS Pathogens.

Best regards,

Richard J. Kuhn, PhD

Academic Editor

PLOS Pathogens

Ana Fernandez-Sesma

Section Editor

PLOS Pathogens

Kasturi Haldar

Editor-in-Chief

PLOS Pathogens

orcid.org/0000-0001-5065-158X

Michael Malim

Editor-in-Chief

PLOS Pathogens

orcid.org/0000-0002-7699-2064

The authors have answered the additional queries listed by the reviewers. This is a very interesting set of data and is suitable for publication.
---

## [Editor Report · Acceptance letter]

5 Jan 2023

Dear Dr. de Silva,

We are delighted to inform you that your manuscript, "Structure and Neutralization Mechanism of a Human Antibody Targeting a Complex Epitope on Zika Virus," has been formally accepted for publication in PLOS Pathogens.

Best regards,

Kasturi Haldar

Editor-in-Chief

PLOS Pathogens

orcid.org/0000-0001-5065-158X

Michael Malim

Editor-in-Chief

PLOS Pathogens

orcid.org/0000-0002-7699-2064